# Margin of Exposure Analyses and Overall Toxic Effects of Alcohol with Special Consideration of Carcinogenicity

**DOI:** 10.3390/nu13113785

**Published:** 2021-10-25

**Authors:** Alex O. Okaru, Dirk W. Lachenmeier

**Affiliations:** 1Department of Pharmacy, University of Nairobi, Nairobi P.O. Box 19676-00202, Kenya; alex.okaru@gmail.com; 2Chemisches und Veterinäruntersuchungsamt (CVUA) Karlsruhe, Weissenburger Straße 3, 76187 Karlsruhe, Germany

**Keywords:** alcohol, risk assessment, hepatotoxicity, dose–response relationship, margin of exposure, epidemiological methods

## Abstract

Quantitative assessments of the health risk of the constituents of alcoholic beverages including ethanol are reported in the literature, generally with hepatotoxic effects considered as the endpoint. Risk assessment studies on minor compounds such as mycotoxins, metals, and other contaminants are also available on carcinogenicity as the endpoint. This review seeks to highlight population cancer risks due to alcohol consumption using the margin of exposure methodology. The individual and cumulative health risk contribution of each component in alcoholic beverages is highlighted. Overall, the results obtained consistently show that the ethanol contributes the bulk of harmful effects of alcoholic beverages, while all other compounds only contribute in a minor fashion (less than 1% compared to ethanol). Our data provide compelling evidence that policy should be focused on reducing total alcohol intake (recorded and unrecorded), while measures on other compounds should be only secondary to this goal.

## 1. Introduction

The epidemiological association of alcoholic beverages with cancer remains a topic that has continued to attract global attention for over a century with the first documented cases, cancer of the esophagus, being reported in 1910 [1,2]. Later in 1988, the World Health Organization (WHO)/International Agency for Research on Cancer (IARC) classified “alcohol drinking” as carcinogenic to humans (group 1) after establishing a causal link between alcohol use and malignancies of the oral pharynx, esophagus, and liver [1]. The promoters or causative factors in alcoholic beverages for developing carcinogenic lesions are a matter of continuing debate among scientists. However, alcohol being a multicomponent mixture, the potential contribution of each or all the compounds to carcinogenesis should not be overlooked. These substances occur as residues, contaminants, or even adulterants, in addition to being naturally occurring in either raw materials or fermentation by-products.

Ethanol, the principal component of alcoholic beverages, is classified as a human carcinogen (group 1) by IARC. Other than ethanol, other IARC-classified carcinogenic compounds such as acetaldehyde, formaldehyde, acrylamide, aflatoxins, ochratoxin A, arsenic, lead, cadmium, ethyl carbamate, furan, safrole, 4-methylimidazole, N-nitrosodimethylamine (NDMA), 3-Monochloropropane-1,2-diol (3-MCPD), and benzene have occurred in alcoholic beverages. The contribution of these compounds to cancer is either synergistic or independent of each other. Understanding the contribution of each component is important in disentangling the mechanisms of carcinogenicity due to alcohol and ultimately aids in alcohol control policies. Nevertheless, epidemiological research has reported that only ethanol achieves the requisite threshold to explain the carcinogenic risk of alcoholic beverages. This review seeks to highlight population cancer risks due to alcohol consumption using the margin of exposure (MOE) approach with emphasis on the cancer-risk contribution of individual components of alcoholic beverages. This review identifies ethanol as the main oncogenic component in alcoholic beverages and lays emphasis on the need for policy geared towards the reduction in drinking per se and not target on other minor carcinogens that may require strict implementation of industry best practices, i.e., as low as reasonably achievable (ALARA) guidelines and good manufacturing practices.

## 2. The Margin of Exposure Method and Its Application to Alcoholic Beverages

Despite there being other methods for evaluating the health risks associated with alcohol intake, the margin of exposure (MOE) method is recommended for comparing the risks of different alcoholic beverage components [1]. MOE compares exposure levels to a toxicological threshold. The toxicological thresholds are derived from the dose–response evaluations for both carcinogens and non-carcinogens.

The ratio between the benchmark dose’s lower one-sided confidence limit (BMDL) and predicted human consumption/exposure of the same substance is known as the margin of exposure. MOE is typically used to compare the health risks of various chemicals and, as a result, to prioritize risk management efforts. The lower the MOE, the greater the risk to people; typically, a value of less than 10,000 is used to indicate health risk.

The benchmark dose (BMD) is the dose of a chemical that, based on the dose–response modeling, causes a specified change in the response rate (benchmark response) of an undesirable impact compared to the background. The benchmark response is typically suggested to be set near the lower limit of what can be measured (e.g., for animal experiment in the 1–10% range). BMD–response modeling results can then be used with exposure data to create a MOE for quantitative risk assessment. No observed effect level (NOEL) or no observed adverse effect level (NOAEL) values can be used as surrogate thresholds where BMDL values are unavailable in the literature. Consequently, the MOEs can be determined by dividing the NO(A)EL by estimated human intake [1].

The human intakes for each beverage group (i.e., beer, wine, spirits, and unrecorded alcohol) for various drinking scenarios (e.g., low risk drinking and heavy drinking) can be based on drinking guidelines such as the Canadian ones, which consider 13.6 g of pure alcohol a standard drink [1]. MOEs for average and maximum contamination with the various substances can also be determined for both drinking scenarios to give a range for average and worst-case contamination scenarios [1].

The most recent detailed IARC reviews were suggested to be used to select compounds and their levels in alcoholic beverages. For the established and probable human carcinogens, toxicological endpoints and BMD are primarily based on literature data [1]. Suitable risk assessment studies, including endpoints and dose–response modeling results, were typically identified in monographs published by national and international risk assessment bodies such as the United States Environmental Protection Agency (US EPA), the World Health Organization International Programme on Chemical Safety (WHO-IPCS), the Joint FAO/WHO Expert Committee on Food Additives (JECFA), and the European Food Safety Authority (EFSA). Data from peer-reviewed scientific research can be used for compounds without accessible monographs or those with missing data on dose–response modeling findings [1].

## 3. Occurrence of Carcinogenic Compounds in Alcoholic Beverages

Ethanol and acetaldehyde (ethanal), both categorized by IARC as group 1 carcinogens, are the primary carcinogens occurring in alcoholic beverages accounting for approximately 5.5% of all cancer cases worldwide [3]. Although the inherent cancer risk of alcoholic beverages parallels consumption volumes, even light alcohol drinking has been associated with cancer with ethanol and acetaldehyde being central to the pathogenicity. At the molecular level, ethanol and acetaldehyde are postulated to cause cancer in similar mechanistic fashion, since acetaldehyde, a genotoxic compound, is a metabolite of ethanol resulting from the alcohol dehydrogenase or CYP 450 E1 pathways. Since ethanol and acetaldehyde have similar carcinogenesis mechanisms, the computation of cancer risk can be be undertaken cumulatively. Additionally, ethanol plays a promoting role in oncogenesis by solvating other carcinogens [1].

Besides metabolism, acetaldehyde occurs naturally, albeit in small amounts in alcoholic beverages with the highest contents reported to be in fortified wines (118 mg/L) and some spirit drinks (66 mg/L) [4]. Additionally, acetaldehyde occurs at high levels in certain unrecorded alcohols [5]. The average daily acetaldehyde exposure from alcoholic beverages has been calculated to be 0.112 mg/kg body weight, with a MOE of 498 [5].

IARC has classified formaldehyde (methanal), a naturally occurring substance found in various plants, mainly fruits and vegetables, and animal products such as meat, dairy products, and fish [1], as a group 1 carcinogen [6]. Formaldehyde is a carcinogen linked to the development of leukemia and naso-pharyngeal cancer in humans. Alcoholic beverages contain a substantial quantity of formaldehyde [7]. In a sampling of 500 beverages including wine, beer, spirits, and unrecorded alcohol, lower formaldehyde contamination (1.8 percent) was found, which was however more than the WHO IPCS permissible concentrations [7]. To surpass the daily US EPA reference dose (RfD) of 0.2 mg/kg bodyweight [8], a person weighing 60 kg would need to partake daily 800 mL of alcohol containing 14.37 mg/L formaldehyde. Even in the worst-case scenario, this level of exposure is exceedingly unlikely.

Acrylamide, considered by IARC as probably carcinogenic, may produce cancer through its metabolite, glycidamide, that forms DNA adducts [9]. Nevertheless, there are only a few reports on the occurrence of acrylamide in alcoholic beverages with one study reporting acrylamide levels of 22 µg/kg [10]. The group 2B carcinogen, 3-monochloropropane-1,2-diol, is a heat-induced contaminant resulting from the thermal processing of malt [11]. In experimental animals, 3-MCPD causes renal tubule adenocarcinomas. Although 3-MCPD is detected in some dark specialty malts used for beer production [11,12,13], it only occurs in low levels in most beers. It typically ranges from <10 μg/L to 14 µg/L [14,15].

IARC has classified the mycotoxins, aflatoxin B_1_ and ochratoxin A, found in some alcoholic beverages as carcinogenic to humans (group 1) and possibly carcinogenic to humans (group 2B), respectively [16]. Aflatoxin B_1_, as well as other aflatoxins (B_2_, G_1_, and G_2_), is a naturally occurring toxin in barley, corn, and sorghum malts that enters beer due to the use of contaminated cereals [17,18,19]. The occurrence of aflatoxins is climate-related with aflatoxins thriving in warm climates, especially in the tropics. Indeed, higher contamination of beer is reported to be in warm climatic countries such as South Africa, India, Mexico, and Kenya, among others [19,20]. Aflatoxin B_1_ has been found in the greatest concentrations (up to 6.8 µg/L) in artisanal beers from Kenya [20,21]. Similarly, ochratoxin A (OTA) occurs as a contaminant in grapes and in raw materials for beer, such as barley, malt, or cereal derivatives. Unlike aflatoxin B_1_, OTA is partially detoxified during fermentation [22], and its concentration remains unchanged in wine for one year [23].

Among heavy metals, arsenic, cadmium, and lead are possibly the ones of carcinogenic concern. The IARC classifies metalloid arsenic and inorganic arsenic compounds as group 1 carcinogens [24]. Lung, skin, liver, kidney, prostate, and urinary bladder malignancies have all been linked to inorganic arsenic compounds [24]. The reported levels of arsenic in beer are 0–102.4 µg/L [24], while those in spirits and wines are 0–27 and 0–14.6 µg/L, respectively [25]. The IARC designated cadmium as a group 1 carcinogenic agent because it causes cancers of the lungs, kidneys, and prostate [26]. According to an EFSA report [25], the amount of arsenic in various beverages varies. Fortified and liqueur wines had a Cd concentration of 0.5 µg/L, whereas liqueur had a level of 6.0 µg/L. The average concentration of Cd in wines and beers is 1.2 and 1.8 µg/L, respectively [25]. Organic lead compounds are “not classifiable as to their carcinogenicity to humans” (group 3) [27], whereas inorganic lead and lead compounds are “probably carcinogenic” (group 2A) [28]. The concentrations of lead vary across alcohol types. The average content of Pb in wines is 29 µg/kg with no significant differences in the amounts between the red and white varieties. Beer and beer-like beverages contain 12 µg/kg Pb on average [29].

Benzene, a heat-induced contaminant, is classified as a group I carcinogen, and it arises in alcoholic beverages. Benzene is a genotoxic compound that targets pluripotent hematopoietic stem cells leading to a raft of chromosomal aberrations [30]. The compound can occur in soft beverages that contain benzoic acid (a preservative) [31,32,33] or in beers manufactured with benzene-contaminated industrial carbon dioxide [34,35].

Furan, a group 2B carcinogen [36], is touted to intercalate with DNA via its cytochrome P-450-mediated metabolite, *cis*-2-butene-1,4-dial [37,38] leading to carcinogenesis. Furan has been found in beer samples at amounts as high as 28 µg/kg. Lower furan concentrations have been reported in wines and liqueurs, 6.5 and 28 µg/kg, respectively [39].

In 2015, IARC classified the controversial herbicide glyphosate as “probably carcinogenic to humans” based on some evidence in humans due to a correlation with non-Hodgkin lymphoma and significant evidence for glyphosate’s carcinogenicity in experimental animals [40]. In 2013, Nagatomi et al. observed that glyphosate content in 15 commercial canned beers from Japan was below the limit of quantitation (10 µg/L) [41]. From a risk assessment standpoint, these observed amounts are unlikely to cause harm.

Ethyl carbamate, a probable human carcinogen (group 2A) [42], has been found in small concentrations in wines and beers (in µg/L) [43] and in larger proportions in stone-fruit spirits (in mg/L) [43]. Another group 2A carcinogenic compound, *N*-nitrosodimethylamine (NDMA), is hepatotoxic [27]. Ethanol through its solvation effect or via alteration of cellular metabolism and suppression of DNA repair, enhances the carcinogenicity of NDMA [44]. NDMA in alcoholic beverages can arise from the manufacturing processes or from storage. During the production process, *N*-nitroso compounds can emerge by activities such as when malt is directly heated or when polluted water is used, or when foods and beverages are stored [45,46]. In a follow-up screening of German beers conducted between 1992 and 2006, NDMA was found in 29 malt samples (43%) and 81 beer samples (7%), with only 4% of the beer samples (*n* = 1242) having concentrations above the technical threshold value [47]. 

Pulegone, a component of essential oil-containing plants of the mint family, is found in mint-flavored alcoholic beverages [48]. Pulegone has been linked to liver and bladder cancer in animal models, prompting the IARC to classify it as probably carcinogenic to humans (group 2B) [48]. Despite being recognized as a potential carcinogen, occurrence data on pulegone are still scanty with only the National Toxicology Programme (NTP) reporting a mean value of 10.5 µg/L [49].

Safrole, a substituted benzodioxole, is a genotoxic agent that naturally occurs in several spices such as sweet basil, black pepper, cinnamon nutmeg, mace, cinnamon, and aniseed. Moreover, safrole occurs in food and beverages that are flavored with it. The IARC categorizes safrole as “possibly carcinogenic to humans” (group 2B) [27]. Since safrole occurs in cola drinks [50], it has the potential to occur in alcoholic beverages [51] especially admixtures of cola and alcohol. On average, humans consume 0.3 mg of safrole per day, with the 97.5th percentile consuming 0.5 mg. The presence of possibly carcinogenic compounds in alcoholic beverages is summarized in Table 1.

## 4. Comparative Risk Assessment of Compounds in Alcoholic Beverages

The presence of a carcinogenic compound in an alcoholic beverage does not directly impute an inherent risk of consumers of the drink. However, the quantitative risk assessment serves to ascribe harm due to a compound if it exceeds the toxicological threshold. The margin of exposure (MOE) methodology as described in the literature is applicable to conduct a comparative risk assessment for compounds in alcoholic beverages [1,5,60,61,62]. Where human data were unavailable, animal data were used instead for risk assessment. Moreover, non-cancer endpoints were chosen for substances such as Pb where there was no dose–response modeling data for cancer effects available. However, non-cancer endpoints may be more sensitive than cancer endpoints. Additionally, the most sensitive endpoint was chosen if dose–response data for several organ sites were available. Table 2 lists the toxicological endpoints and points of departure used in dose–response modeling and risk assessment. 

For daily consumption of four standard alcoholic drinks, MOEs were calculated for the average and worst-case scenarios. For ethanol, the lowest MOE was achieved (0.8). Inorganic lead and arsenic showed MOEs ranging from 10 to 300, while acetaldehyde, cadmium, ethyl carbamate, and pulegone had MOEs ranging from 1000 to 10,000. Safrole, ochratoxin A, NDMA, 4-methylimidazole, 3-MCPD, glyphosate, furan, formaldehyde, and acrylamide had average MOEs exceeding 10,000 even in these extreme contexts such as binge drinking (Figure 1). However, the MOE for aflatoxin B_1_ from Kenyan artisanal beer that was significantly tainted ranged from 15 to 58 with a mean of 36. As a result, ethanol is the most significant carcinogen found in alcoholic beverages, with a clear dose–response relationship. Other contaminants (lead, arsenic, ethyl carbamate, acetaldehyde) may pose risks below those tolerated for food contaminants, but from a cost-effectiveness standpoint, the focus should be on reducing alcohol consumption in general rather than on mitigative actions for some contaminants that contribute only a small (if any) portion of the total health risk. This review again highlights the fact that ethanol remains the compound with the highest carcinogenic potential that is present in alcoholic beverages. This finding is consistent with other studies reported in the literature [1,21,60,62]. Aflatoxin B_1_ also emerged as a compound of interest in unrecorded artisanal beers that clearly requires attention in the warm climatic countries where the consumption of such beers is prevalent [20,21]. Figure 1 shows the comparative MOEs for carcinogens.

## 5. Overall Toxic Effects of Alcoholic Beverages

According to studies, no amount of alcohol use promotes health [94]. Alcohol consumption significantly contributes to death, disability, and ill health worldwide [94,95,96]. Alcohol is the sixth most common cause of mortality and disability-adjusted life years (DALYs) in both men and women, accounting for 22% of female fatalities and 68% of male deaths [94]. There is a link between harmful alcohol consumption and various mental and behavioral illnesses, as well as other non-communicable diseases such as tuberculosis and HIV/AIDS and injuries. Injuries constitute the greatest negative consequence of alcohol consumption after cancer. Cardiovascular disease accounts for 15% of alcohol-attributable morbidity, while liver cirrhosis accounts for 13% of all alcohol-attributable deaths [97]. Besides the health risks, irresponsible alcohol use results in social and economic losses for consumers and the community as a whole [98,99].

## 6. Conclusions

Despite there being other methods for evaluating the health risks associated with alcohol intake, the margin of exposure method is recommended for comparing the risks of different alcoholic beverage components. From this review, ethanol remains the most prominent carcinogen in alcoholic beverages, according to quantitative comparative risk assessment. Therefore, the reduction in alcohol intake ought to be prioritized in combating harm due to alcoholic beverages [100]. Since the dose–response relationship holds for alcohol harm, reduction in alcoholic strength would be beneficial in minimizing the harmful effects of alcohol [101]. For illustration, drinking four bottles of 5.5 percent vol. ethanol beer generates a MOE of 0.5, whereas drinking the same volume of light beer (1.5 percent vol. ethanol) yields a substantially greater MOE of 1.9 [1]. Moreover, consumers may not be typically able to discriminate different alcohol strengths in beer and, thus, may not ingest more volumes to compensate for the lower alcoholic strength beer [102,103].

Other carcinogens besides ethanol require mitigative steps as well, which may require strict adoption of industry best practices such as keeping contaminants/components as low as can reasonably be achieved (ALARA). We urge the relevant regulatory authorities to implement the available mitigative measures to protect consumers from potentially carcinogenic substances.

## Figures and Tables

**Figure 1 nutrients-13-03785-f001:**
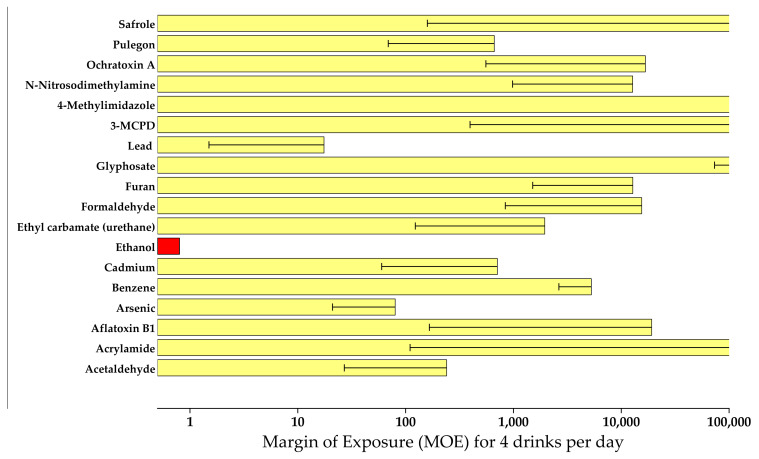
Comparative MOEs for IARC-classified carcinogens in alcoholic beverages (reprinted with permission from Springer Nature, Archives of Toxicology, Pflaum et al. [1], copyright 2016).

**Table 1 nutrients-13-03785-t001:** Summary of the occurrence of potentially carcinogenic compounds in alcoholic beverages (reprinted with modifications with permission from Springer Nature, Archives of Toxicology, Pflaum et al. [1], copyright 2016).

Agent (IARC Group ^a^)	Beverage Type	Concentration		Reference
		Average	Maximum	
Acetaldehyde in alcoholic beverages (1)	Beer	9 mg/L	63 mg/L	[4]
	Spirit	66 mg/L	1159 mg/L	
	Wine	34 mg/L	211 mg/L	
Acrylamide ^b^ (2A)	Beer	0–72 µg/kg	363 µg/kg	[15,52]
Aflatoxins (1)	Commercial beer	0.002 µg/L	0.230 µg/L	[17]
	Artisanal beer	3.5 µg/L	6.8 µg/L	[20]
Arsenic (1)	Beer	0 µg/L	102.4 µg/L	[1]
	Spirit	13 µg/L	27 µg/L	
	Wine	13 µg/L	27 µg/L	
Benzene (1)	Beer	10 µg/L	20 µg/L	[1]
Cadmium (1)	Beer	0.9 µg/L	14.3 µg/L	[1]
	Spirits	6 µg/L	40 µg/L	
	Wine	1.0 µg/L	30 µg/L	
Ethanol (1)	Varies	2% vol.	80% vol.	[1]
Ethyl carbamate (2A)	Beer	0 µg/kg	33 µg/kg	[53]
	Spirits	93 µg/kg	6730	
	Stone spirits	744 µg/kg	22,000 µg/kg	
	Wine	5 µg/kg	180 µg/kg	
Formaldehyde (1)	Beer	0 mg/L	0 mg/L	[8]
	Spirits	0.50 mg/L	14.37 mg/L	
	Wine	0.13 mg/L	1.15 mg/L	
Furan (2B)	Beer	3.3 µg/kg	28 µg/kg	[39]
Glyphosate ^c^ (2A)	Beer	0–30 µg/L		[1]
Lead compounds, inorganic (2A)	Beer	2 µg/L	15 µg/L	[54]
	Spirits	31 µg/L	600 µg/L	[1]
	Wine	57 µg/L	236 µg/L	[55]
MCPD ^d^ (2B)	Beer	0–14 µg/kg		[12]
4-Methylimidazole ^e^ (2B)	Beer^e^	9 µg/L	28 µg/L	[56]
	Spirit	0 µg/L	0.014 µg/L	[57]
NMDA (2A)	Beer	0.1 µg/kg	1.3 µg/kg	[1]
Ochratoxin A (2B)	Beer	0.05 µg/L	1.5 µg/L	[1]
	Wine	0.23 µg/L	7.0 µg/L	
Pulegone ^f^ (2B)		10.5 mg/kg	100 mg/kg	[49,58]
Safrole (2B)	Liqueurs, aperitifs, and bitters	ND	6.6 mg/L	[51]

Abbreviations: MCPD—3-Monochloropropane-1,2-diol, NMDA—N-Nitrosodimethylamine, ND—below the limit of quantitation. ^a^ Only compounds present in alcoholic beverages that fall under IARC groups 1 (carcinogenic to humans), 2A (probably carcinogenic to humans), and 2B (possibly carcinogenic to humans) were included in this list. ^b^ There are few studies on acrylamide in alcoholic beverages. Most samples examined had levels that were below the detection limit. A single sample of wheat beer had a level of 72 µg/kg, while craft beers found in Poland and the Czech Republic had 363 µg/kg [52]. ^c^ Except for the “worst-case” scenario, upper level of 30 µg/L [59] was used, since there is a dearth of systematic data on the occurrence glyphosate in beer. ^d^ There was limited research on the presence of 3-MCPD in alcoholic beverages. As a result, the upper limit was set at less than 10 µg/L from a study on beers [12]. ^e^ Caramelized alcoholic beverages. ^f^ Studies on the occurrence of pulegone in alcoholic beverages are scanty. Thus, 10.5 mg/kg [49] and 100 mg/kg [58] were utilized as the minimum and maximum amounts of pulegone, respectively, in alcohol products.

**Table 2 nutrients-13-03785-t002:** Dose–response modeling for potential human carcinogens occurring in alcoholic beverages (reprinted with permission from Springer Nature, Archives of Toxicology, Pflaum et al. [1] copyright 2016).

Carcinogenic Agent	Modeling Toxicological Endpoint	Animal Model	Route/Mode of Exposure	BMDL ^a^	
				(mg/kg bw/Day)	Reference
Acetaldehyde	Animal tumors [60]	Male rats	Oral	56	[60]
Acrylamide	Harderian gland tumors [63]	Mice	Oral	0.18	[64]
Aflatoxin B_1_	Cancer of the lungs in humans [65]	NA	Food	0.00087	[66]
Arsenic	Cancer of the lungs in humans [67]	NA	Water	BMDL_0.5_: 0.003	[68]
Benzene	Human lymphocyte count [69]	NA	Inhalation extrapolated to oral	1.2 ^b^	[70]
Cadmium	Human studies [70]	NA	Food	NOAEL: 0.01 ^c^	[70]
Ethanol	Hepatocellular adenoma or carcinoma [71]	Rats	Oral	700	[72,73]
Ethyl carbamate	Bronchiolar alveolar carcinoma [74]	Mice	Oral	0.3	[73]
Formaldehyde	The aerodigestive tract, comprising the oral and gastrointestinal mucosa, undergoes histological alterations [75]	Rats	Oral	NOEL: 15 ^c^	[76]
Furan	Adenomas and carcinomas of the liver [77]	Female mice	Oral	0.96	[78]
Glyphosate ^b^	There are no dose–response data for the cancer outcome			NOAEL: 50	[79]
Lead	Human cardiovascular effects [29]	NA	Diet	BMDL10: 0015 ^d^	[80]
3-MCPD	Hyperplasia of the tubules of the kidneys ^e^ [81]	Rats	Oral	0.27	[82]
4-Methylimidazole	Lung cancer [83]	Mice	Oral	NOAEL: 80 ^c^	[84]
*N*-Nitrosodimethylamine	Hepatocellular carcinoma [85]		Oral	0.029	[86,87]
Ochratoxin A	Renal adeno-carcinoma [88]	Male rats	Oral	0.025	[89]
Pulegone	Urinary bladder tumors [90]	Rats	Oral	LOAEL: 20 ^c^	[49]
Safrole	Hepatic tumors [91]	Mice	Oral	3 ^f^	[92,93]

NA—not applicable. ^a^ For an x % occurrence of health effect, BMDLx is the lower one-sided confidence limit of the benchmark dose (BMD). ^b^ Inhalation exposure was used as the original endpoint. Route-to-route extrapolation was used to calculate the BMDL for oral exposure [69]. ^c^ The no effect level (NOEL), no observed adverse effect level (NOAEL), or lowest observed adverse effect level (LOAEL) were utilized because no appropriate BMD modeling for exposure through the mouth has been documented. ^d^ Overall exposure to lead is determined in blood, and the figures are based on that. The BMDL that was employed was determined based on dietary exposure [29]. ^e^ Renal tubular hyperplasia, rather than renal tubule adenoma or cancer, was a more sensitive endpoint. ^f^ This was a conservative minimal concentration based on the literature’s BMDL10 range of “about 3–29 mg/kg bw/day” for safrole [91].

## Data Availability

Extracted data are presented in the main tables.

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
