# Peer review of "Margin of Exposure Analyses and Overall Toxic Effects of Alcohol with Special Consideration of Carcinogenicity"

_nutrients, 2021, doi:10.3390/nu13113785_

Round 1

Reviewer 1 Report

The manuscript entitled „Margin of exposure analyses and overall toxic effects of alcohol with special consideration of carcinogenicity” .

General comments

  • The Introduction needs a lot of improvement. Table 1 in this chapter should be deleted.
  • Authors did not present the purpose of the work.
  • Authors did not provide a justification for the issue and the usefulness of the issue.
  • I do not see sense in determining the toxicity of individual substances present in alcoholic beverages for many reasons:
  • firstly, alcoholic beverages are characterized by different nutritional value and ethanol content resulting from the raw material used, production method, etc. For this reason, we distinguish many types of beer or wine.
  • secondly, we drink the alcoholic beverages as a whole, it is not possible to separate ethanol from other substances, so determining their toxicity separately seems to be of little use.
  • thirdly, it is known that the most harmful substance is ethanol and acetaldehyde.
  • Authors provide the characteristics of substances found in alcoholic beverages, which is nothing new or revealing.
  • In Conclusion section Authors state something that is well known and does not constitute new knowledge about the health consequences of consuming alcoholic beverages.
  • It will be better if this manuscript was a meta-analysis or systematic review with a well-described methodology.
  • The presented manuscript does not constitute a new approach to the topic.

Author Response

The Introduction needs a lot of improvement. Table 1 in this chapter should be deleted.

RESPONSE: The introduction as well as the whole text was carefully revised. Table 1 was deleted and some information implemented into table 2.

Authors did not present the purpose of the work.

Authors did not provide a justification for the issue and the usefulness of the issue.

RESPONSE: We have now included the aim of this review for clarity and added a justification statement at the end introductory section of this paper.

I do not see sense in determining the toxicity of individual substances present in alcoholic beverages for many reasons:

firstly, alcoholic beverages are characterized by different nutritional value and ethanol content resulting from the raw material used, production method, etc. For this reason, we distinguish many types of beer or wine.

secondly, we drink the alcoholic beverages as a whole, it is not possible to separate ethanol from other substances, so determining their toxicity separately seems to be of little use.

thirdly, it is known that the most harmful substance is ethanol and acetaldehyde.

Authors provide the characteristics of substances found in alcoholic beverages, which is nothing new or revealing.

RESPONSE: We believe that the risk assessment of individual compounds has some merit, not only from a consumer’s perspective, but also from a scientific perspective, as there is still much misinformation regarding the individual contribution of risk. The alcohol industry tend to exaggerate the risk of the individual compounds, to focus policy measures on these, minor issues, but not on the larger health issues, namely the reduction of drinking per se.

Regarding the industry influence as well as an up-to-date review of compounds in unrecorded alcohol, we would like to refer to the following references. In the context of the current paper, we did not want to expand on these issues which are not the point of the review.

Rehm, J., Neufeld, M., Room, R., Sornpaisarn, B., ŠtelemÄ—kas, M., Swahn, M. H., & Lachenmeier, D. W. (2021). The impact of alcohol taxation changes on unrecorded alcohol consumption: a review and recommendations. International Journal of Drug Policy, 103420.

Lachenmeier, D. W., Neufeld, M., & Rehm, J. (2021). The impact of unrecorded alcohol use on health: What do we know in 2020?. Journal of Studies on Alcohol and Drugs82(1), 28-41.

Finally, we want to point out that the topic of the paper was based on a specific invitation to this special issue of Nutrients, and we just delivered the review of the topic the journal editor asked for. For this reason alone, there appears to be some “sense” as well as public interest in determining the toxicity of individual substances present in alcoholic beverages.

In Conclusion section Authors state something that is well known and does not constitute new knowledge about the health consequences of consuming alcoholic beverages.

It will be better if this manuscript was a meta-analysis or systematic review with a well-described methodology.

The presented manuscript does not constitute a new approach to the topic.

RESPONSE: We basically agree with this assessment. However, we were asked for a review about the topic and not an original research manuscript. A review paper can never provide “new knowledge” when it is tasked to provide an overview of the current state of knowledge. To be transparent, there was not much progress since our original research papers on the topic. However, as there is so many misinformation surrounding this topic, this update about MOE analyses of alcoholic beverages should be still worthwhile to being published.

The author guidelines for reviews specify: “Reviews: These provide concise and precise updates on the latest progress made in a given area of research“. We believe that we have delivered this. There is no requirement for novelty or impact as in classical print journals.

Reviewer 2 Report

This review highlited the cancer risks due to alcohol consumption using the MOE. severy typos need to be corrected such as line 122 "through is metabolite".

Author Response

This review highlited the cancer risks due to alcohol consumption using the MOE. severy typos need to be corrected such as line 122 "through is metabolite".

RESPONSE: Thank you for pointing out this mistake. We have carefully copy-edited the manuscript.

Round 2

Reviewer 1 Report

Thank you for correcting the manuscript.

This manuscript is a resubmission of an earlier submission. The following is a list of the peer review reports and author responses from that submission.